# In situ architecture of neuronal α-Synuclein inclusions

Victoria A. Trinkaus[1,2,3], Irene Riera-Tur [4,5], Antonio Martínez-Sánchez [6,7,8], Felix J. B. Bäuerlein [6,7,8], Qiang Guo[6,9,10,11], Thomas Arzberger[3,12,13], Wolfgang Baumeister[6], Irina Dudanova [4,5], Mark S. Hipp [1,3,14,15], F. Ulrich Hartl [1,3,16 ✉] & Rubén Fernández-Busnadiego [6,7,8,16 ✉]

The molecular architecture of α-Synuclein (α-Syn) inclusions, pathognomonic of various neurodegenerative disorders, remains unclear. α-Syn inclusions were long thought to consist mainly of α-Syn fibrils, but recent reports pointed to intracellular membranes as the major inclusion component. Here, we use cryo-electron tomography (cryo-ET) to image neuronal α-Syn inclusions in situ at molecular resolution. We show that inclusions seeded by α-Syn aggregates produced recombinantly or purified from patient brain consist of α-Syn fibrils crisscrossing a variety of cellular organelles. Using gold-labeled seeds, we find that aggregate seeding is predominantly mediated by small α-Syn fibrils, from which cytoplasmic fibrils grow unidirectionally. Detailed analysis of membrane interactions revealed that α-Syn fibrils do not contact membranes directly, and that α-Syn does not drive membrane clustering. Altogether, we conclusively demonstrate that neuronal α-Syn inclusions consist of α-Syn fibrils intermixed with membranous organelles, and illuminate the mechanism of aggregate seeding and cellular interaction.

[1] Department of Cellular Biochemistry, Max Planck Institute of Biochemistry, Martinsried, Germany. [2] Graduate School of Quantitative Biosciences Munich, Munich, Germany. [3] Munich Cluster for Systems Neurology (SyNergy), Munich, Germany. [4] Molecular Neurodegeneration Group, Max Planck Institute of Neurobiology, Martinsried, Germany. [5] Department of Molecules - Signaling – Development, Max Planck Institute of Neurobiology, Martinsried, Germany. [6] Department of Molecular Structural Biology, Max Planck Institute of Biochemistry, Martinsried, Germany. [7] Institute of Neuropathology, University Medical Center Göttingen, Göttingen, Germany. [8] Cluster of Excellence "Multiscale Bioimaging: from Molecular Machines to Networks of Excitable Cells" (MBExC), University of Göttingen, Göttingen, Germany. [9] School of Life Sciences, Peking University, Beijing, China. [10] Peking-Tsinghua Center for Life Sciences, Peking University, Beijing, China. [11] State Key Laboratory of Protein and Plant Gene Research, Peking University, Beijing, China. [12] Center for Neuropathology and Prion Research, Ludwig-Maximilians-University Munich, Munich, Germany. [13] Department of Psychiatry and Psychotherapy, University Hospital, Ludwig-Maximilians-University Munich, Munich, Germany. [14] Department of Biomedical Sciences of Cells and Systems, University Medical Center Groningen, University of Groningen, Groningen, The Netherlands. [15] School of Medicine and Health Sciences, Carl von Ossietzky University Oldenburg, Oldenburg, Germany. [16] Aligning Science Across Parkinson's (ASAP) Collaborative Research Network, Chevy Chase, MD, USA. ✉email: uhartl@biochem.mpg.de; ruben.fernandezbusnadiego@med.uni-goettingen.de

α-Synuclein (α-Syn) aggregation is a hallmark of several devastating neurodegenerative disorders, including Parkinson's disease (PD) and multiple systems atrophy (MSA)[1,2]. α-Syn aggregates undergo spreading throughout the brain during disease progression[1–3], suggesting mechanisms of intercellular seeding. Similar to other disease-related protein aggregates, pathological α-Syn is thought to adopt the amyloid fold[4]. Formation of α-Syn amyloid fibrils is observed in vitro[5,6] and fibrillar α-Syn has been purified from patient brains[7,8]. Early electron microscopy (EM) studies suggested that Lewy bodies[9–12] and glial cytoplasmic inclusions[13,14], characteristic of PD and MSA, respectively, are fibrillar. However, conventional EM lacks the resolution to unequivocally determine the molecular identity of these fibrils in situ.

Intriguingly, X-ray diffraction measurements confirmed the presence of amyloid fibrils only in some Lewy bodies[15]. A recent study using correlative EM on chemically fixed PD brain tissue suggested that cellular membranes were the main component of Lewy bodies, alongside unidentified fibrillar material[16]. These findings resonated with reports[17] that native α-Syn binds lipids, such as synaptic vesicle membranes[18], observations that lipids can catalyze α-Syn aggregation in vitro[19], and that α-Syn expression in cells is associated with membrane abnormalities[20]. Thus, the disease relevance of fibrillar (amyloid-like) α-Syn aggregation has been questioned, leading to a model in which the main role of α-Syn in pathological inclusions is to cluster cellular membranes[20,21].

Cryo-electron tomography (cryo-ET) is ideally suited to test these new ideas, as it can reveal the molecular architecture of protein aggregates at high resolution within neurons pristinely preserved by vitrification[22–24]. Here, we apply cryo-ET to analyze the fine architecture and cellular interactions of neuronal α-Syn inclusions in situ. We show that α-Syn inclusions consist of α-Syn fibrils intermixed with cellular organelles. However, α-Syn does not link organelles to each other and fibrils do not interact with membranes directly. Furthermore, experiments with gold-labeled extracellular seeds demonstrate that small fibrils are the most seeding-competent species in our preparations.

## Results and discussion
**Neuronal α-Syn inclusions contain α-Syn fibrils.** We performed cryo-ET on neuronal α-Syn aggregates using a well-established seeding paradigm that recapitulates interneuronal spreading and key neuropathological features of Lewy bodies, including the ability to bind amyloid dyes[1,25–27]. Primary mouse neurons were cultured on EM grids, transduced with GFP-α-Syn and incubated with recombinant α-Syn preformed fibrils (PFFs; Supplementary Fig. 1a, b). Unless otherwise stated, all experiments were carried out using the familial A53T α-Syn mutation due to its higher seeding potency[28]. As reported, seeding of neurons led to the formation of GFP-α-Syn inclusions that were positive for Lewy body markers, including phospho-α-Syn (Ser129) and p62 (Supplementary Fig. 1c, d). GFP-α-Syn inclusions in cell bodies or neurites were targeted for cryo-ET by correlative microscopy and cryo-focused ion beam (cryo-FIB) milling[22–24,29,30] (Supplementary Fig. 2). In all cases, this analysis revealed large fibrillar accumulations at sites of GFP-α-Syn fluorescence (Fig. 1a, d and Supplementary Movie 1). Interestingly, the fibrils appeared to be composed of a core decorated by globular GFP-like densities (Fig. 1b), reminiscent of GFP-labeled polyQ and *C9orf72* poly-GA aggregates[22,23]. The fibrils were clearly distinct from cytoskeletal elements (Fig. 1c). However, in contrast to polyQ and poly-GA inclusions, GFP-α-Syn inclusions were populated by numerous cellular organelles, including endoplasmic reticulum, mitochondria, autophagolysosomal structures, and small vesicles (Fig. 1a, d

and Supplementary Movie 1). Thus, the α-Syn inclusions formed in our cellular system recapitulated key molecular and ultrastructural features of PD Lewy bodies and mature Lewy body-like inclusions in culture[12,16,31].

To further investigate the nature of the fibrils observed at sites of GFP-α-Syn fluorescence and avoid possible artifacts caused by GFP-α-Syn overexpression, we next imaged inclusions formed by endogenous α-Syn in neurons seeded by recombinant PFFs. Given the high p62 signal observed in Lewy bodies[1,32] and GFP-α-Syn inclusions (Supplementary Fig. 1d), we expressed p62-RFP as a surrogate marker[23] of endogenous α-Syn inclusions (Supplementary Fig. 1e) to guide correlative cryo-FIB/ET analysis. Although endogenous α-Syn inclusions were smaller than those formed by GFP-α-Syn (Supplementary Fig. 1c), cryo-ET imaging revealed a similar nanoscale organization, consisting of various organelles crisscrossed by abundant fibrils (Fig. 1e, h). Importantly, the fibrils appeared identical to those observed in GFP-α-Syn inclusions (Fig. 1b), except that they were not decorated by globular densities (Fig. 1f). The fibrils were ~10 nm in diameter, similar to recombinant and patient-derived α-Syn fibrils[33,34] and clearly distinct from neurofilaments (Fig. 1g). These data conclusively demonstrate that the fibrils observed in α-Syn inclusions are formed by α-Syn, and argue against a major effect of GFP-α-Syn overexpression on inclusion architecture. Nevertheless, GFP-α-Syn overexpression enhanced the rate of inclusion formation and neuronal toxicity (Supplementary Fig. 1f, g), implicating aggregated α-Syn in neuronal death[27].

Recent studies have demonstrated that amyloid fibrils, including those formed by α-Syn, may adopt different conformations when purified from patient brain in comparison to fibrils generated in vitro from recombinant proteins[33,35]. Therefore, to assess the disease relevance of our findings using recombinant PFFs, we seeded primary neurons expressing GFP-α-Syn with α-Syn aggregates purified from MSA patient brain (Supplementary Fig. 3). Similar to PFFs, MSA seeds triggered the formation of intracellular GFP-α-Syn inclusions positive for phospho-α-Syn (Ser129) and p62 (Supplementary Fig. 3e). Importantly, cryo-ET analysis showed that MSA-seeded neuronal aggregates also consisted of a dense meshwork of α-Syn fibrils interspersed by cellular organelles (Fig. 2a–c). Therefore, our results show that neuronal α-Syn aggregates seeded by MSA patient material are formed by accumulations of α-Syn fibrils intermixed with cellular membranes.

We further investigated possible morphological differences between fibrils seeded by PFFs and MSA aggregates, and in neurons expressing endogenous α-Syn or GFP-α-Syn. In all cases, mean fibril length was ~250 nm (Fig. 2d and Supplementary Table 1). However, fibril density within inclusions, defined as the fraction of cytosolic volume occupied by fibrils, was significantly higher in cells expressing GFP-α-Syn (Fig. 2e and Supplementary Table 1). This was likely due to the higher expression level of this construct, resulting in a higher aggregate load (Supplementary Fig. 1c, f). We next calculated the persistence length of the fibrils to investigate their mechanical properties. Interestingly, whereas PFF-seeded fibrils in neurons expressing GFP-α-Syn or endogenous α-Syn were almost identical in persistence length (Supplementary Fig. 4), MSA-seeded GFP-α-Syn fibrils displayed a considerably lower persistence length (Supplementary Fig. 4), reflecting higher structural flexibility. These values are in the range of those measured for α-Syn[36] and tau[37] fibrils in vitro, as well as for polyQ fibrils in situ[22]. Our measurements are also consistent with single-particle studies reporting a higher twist, indicative of higher flexibility[38], for MSA-derived fibrils[33] (~60 nm) than for most structures of recombinant fibrils[34] (90–120 nm). Thus, different types of exogenous α-Syn aggregates seed neuronal inclusions with different mechano-physical properties.

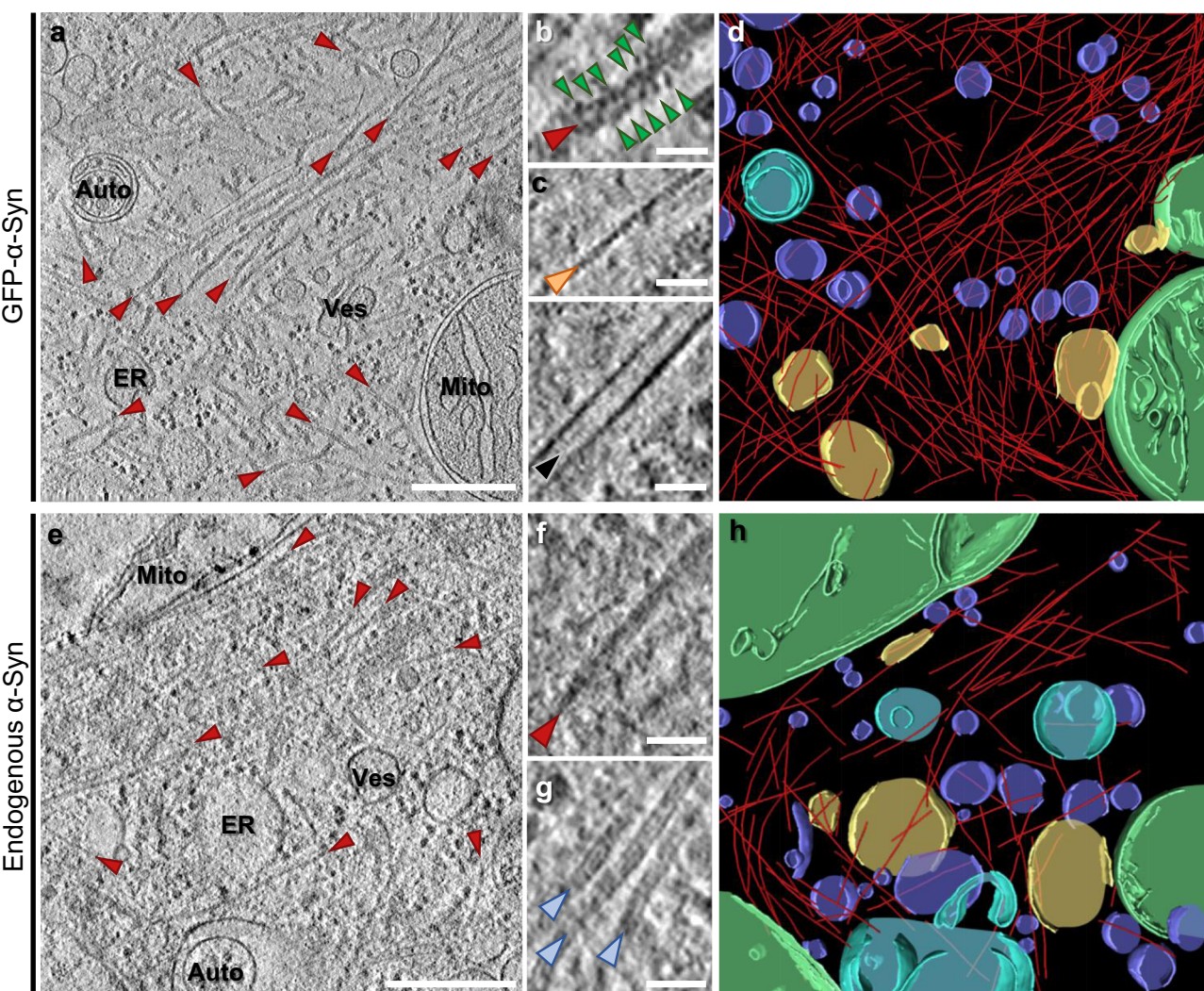

**Fig. 1 Cryo-ET imaging of α-Syn aggregates seeded by PFFs in neurons. a** A tomographic slice (thickness 1.8 nm) of an inclusion seeded by PFFs in a neuron expressing GFP-α-Syn. Auto; autophagosome, ER; endoplasmic reticulum, Mito; mitochondrion, Ves; vesicles. Fibrils are marked by red arrowheads. Scale bar: 350 nm. **b** Magnified view of a fibril with GFP-like densities (green arrowheads) decorating the fibril core. Scale bar: 30 nm. **c** Magnified views of an actin filament (orange arrowhead) and a microtubule (black arrowhead). Scale bars: 30 nm. **d** 3D rendering of the tomogram depicted in **a** showing α-Syn fibrils (red), an autophagosome (cyan), ER (yellow), mitochondria (green), and various vesicles (purple). **e** A tomographic slice (thickness 1.4 nm) of an inclusion seeded by PFFs in a neuron expressing p62-RFP. Scale bar: 350 nm. **f** Magnified view of a fibril. Note that fibrils in cells not expressing GFP-α-Syn are not decorated by GFP-like densities. Scale bar: 30 nm. **g** Magnified view of neurofilaments (blue arrowheads). Scale bar: 30 nm. **h** 3D rendering of the tomogram depicted in **e**. The number of tomograms and biologically independent cryo-ET experiments is listed in Supplementary Table 1. Representative images are shown.

**Small fibrils drive seeding of neuronal α-Syn inclusions**. The seeding of intracellular aggregation by extracellular aggregates may underlie the spreading of pathology across different brain regions during the progression of various neurodegenerative diseases, including synucleinopathies[1–3,39]. To gain a better mechanistic understanding of the seeding process, we tracked the fate of extracellular gold-labeled α-Syn seeds upon internalization into neurons expressing GFP-α-Syn. In this case, we used WT PFFs as they allowed higher labeling efficiency. Recombinant WT α-Syn fibrils were conjugated to 3-nm gold beads via NHS ester coupling, resulting in densely gold-labeled PFFs (Fig. 3a) that efficiently seeded the formation of neuronal GFP-α-Syn inclusions (Supplementary Fig. 5a). Some of these experiments were also carried out in a SH-SY5Y cell line stably expressing GFP-α-Syn as a simpler model system (Supplementary Fig. 6). Interestingly, cryo-ET analysis of inclusions seeded by gold-labeled PFFs showed GFP-α-Syn fibrils with one end decorated by three to ten

gold particles (Fig. 3b, c), indicating that exogenous seeds triggered the fibrillation of cellular α-Syn in a polarized manner, consistent with the polar structures of recombinant α-Syn fibrils[34]. Although patient-derived α-Syn fibrils are polar as well[7,8,33,40], it remains to be established whether disease-related seeds also trigger unidirectional fibril growth in cells. Our data also show that, in our experimental conditions, small α-Syn fibrils are the most seeding-competent species, in agreement with previous results[41]. Therefore, despite the presence of abundant large fibrils in the exogenously added PFF material (Fig. 3a), these species are apparently not efficiently internalized. On the other hand, given the mechano-physical differences between neuronal fibrils growing from PFFs and MSA seeds (Supplementary Fig. 4), the seeding-competent species likely contain the necessary information to confer these structural features. Although α-Syn strains can be transmitted between cells in vitro and in vivo[3], the cellular environment may also modify the strain characteristics[42].

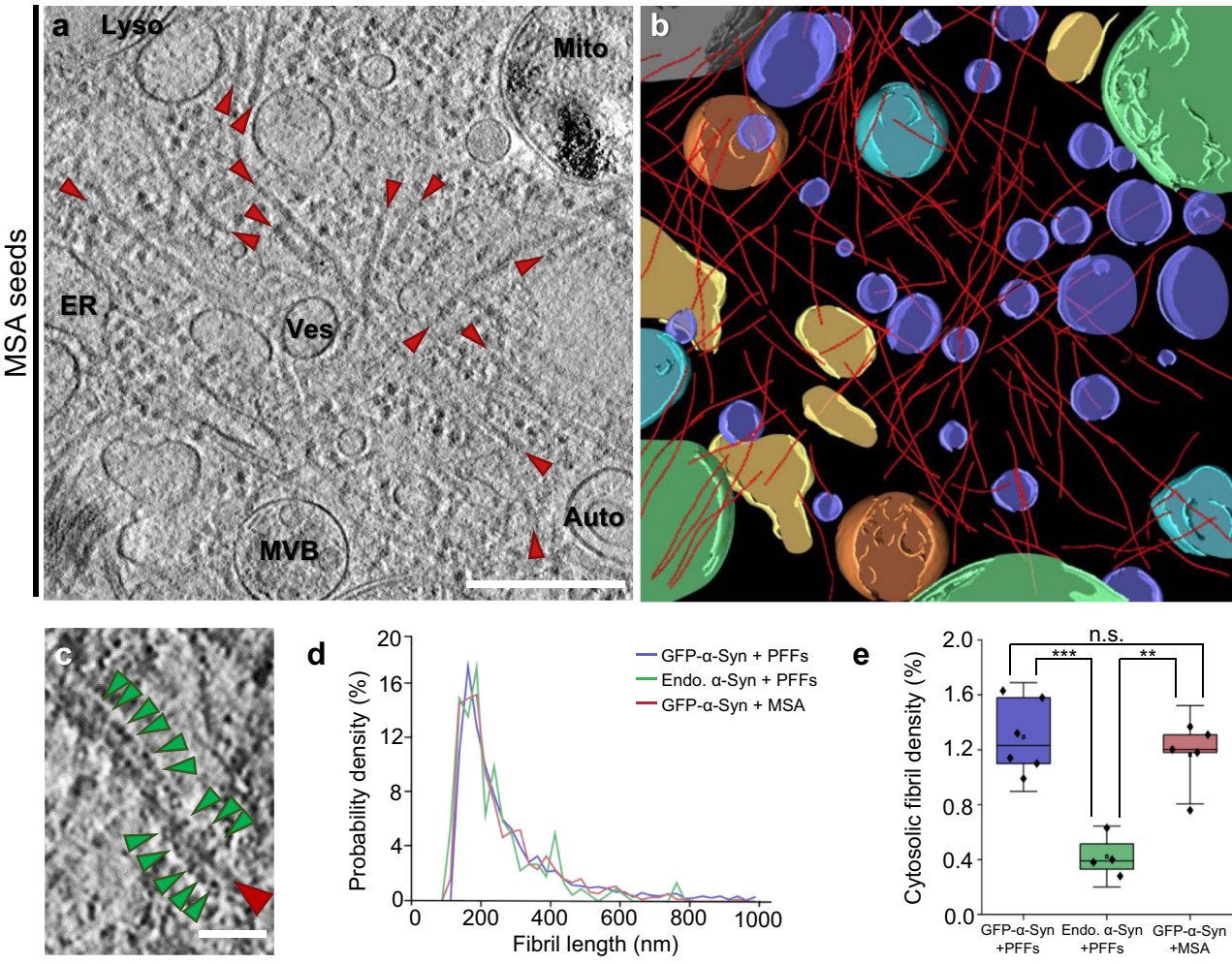

**Fig. 2 Cryo-ET imaging of α-Syn aggregates seeded by MSA patient brain material in neurons. a** A tomographic slice (thickness 1.4 nm) of an inclusion seeded by MSA patient brain material in a neuron expressing GFP-α-Syn. Auto; autophagosome, ER; endoplasmic reticulum, Lyso; lysosome, Mito; mitochondrion, MVB; multivesicular body, Ves; vesicles. Fibrils are marked by red arrowheads. Scale bar: 350 nm. **b** 3D rendering of the tomogram depicted in **a** showing α-Syn fibrils (red), autophagosomes (cyan), ER (yellow), a lysosome (gray), mitochondria (green), multivesicular bodies (orange), and various vesicles (purple). **c** Magnified view of a fibril with GFP-like densities (green arrowheads) decorating the fibril core. Scale bar: 30 nm. **d** Histogram of fibril length. $n = 1592$ (GFP-α-Syn + PFFs), 220 (endogenous α-Syn + PFFs), and 721 (GFP-α-Syn + MSA) fibrils analyzed over three biologically independent experiments for all conditions. **e** Box plots of cytosolic fibril density within inclusions, defined as the fraction of cytosolic volume occupied by fibrils. The horizontal lines of each box represent 75% (top), 50% (middle), and 25% (bottom) of the values, and a black square the average value. Whiskers represent 1.5× standard deviation and black diamonds the individual data points. $n = 6$ (GFP-α-Syn + PFFs), 4 (endogenous α-Syn + PFFs), and 5 (GFP-α-Syn + MSA) tomograms analyzed over three biologically independent experiments for all conditions; n.s., ** and *** indicate respectively $p = 0.4$, $p = 0.0010$, and $p = 4 \times 10^{-4}$ by one-way ANOVA. The number of fibrils, tomograms, and biologically independent cryo-ET experiments is listed in Supplementary Table 1. Representative images are shown in **a–c**. Source data for **d** and **e** are provided as a Source data file.

Thus, higher resolution data are needed to elucidate to what extent the structure of the seed is templated in the aggregates seeded within cells.

Gold-labeled α-Syn was also observed within the lumen of endolysosomal compartments (Fig. 3d and Supplementary Fig. 5b) and at their membrane (Fig. 3e and Supplementary Fig. 5b). Although the growth of α-Syn fibrils was occasionally observed directly at such membrane-bound gold-labeled structures (Fig. 3e, f), most gold-labeled fibrils were cytosolic (Fig. 3b). These data are in agreement with a model where small α-Syn fibrils entering the cell are targeted to endosomes, from which they escape and trigger intracellular growth of α-Syn fibrils[3,43].

**α-Syn does not cluster cellular membranes within inclusions.**
The affinity of α-Syn for lipids[17] has led to the proposal that α-Syn drives the accumulation of cellular membranes in Lewy

bodies[16,20,21], e.g., by fibril-membrane contacts as observed for polyQ fibrils[22]. Such contacts existed within α-Syn inclusions (Supplementary Fig. 7a, b), but they were extremely rare and did not seem to cause the kind of membrane deformations (Supplementary Fig. 7a) seen with polyQ[22]. Although we found a few examples of fibrils contacting membranes at areas of high curvature (Supplementary Fig. 7b), such areas also existed in the absence of fibril contacts (Supplementary Fig. 7c). Thus, apparent fibril-membrane contacts seemed to be mainly a consequence of the crowded cellular environment. To test this hypothesis, we computationally introduced random shifts and rotations to the experimentally determined positions of α-Syn fibrils within the tomograms. This analysis revealed that close fibril-membrane distances (<20 nm) were significantly more frequent in random simulations than in the experimental data (Fig. 4a, b, Supplementary Fig. 7d, and Supplementary Table 1). Together, these

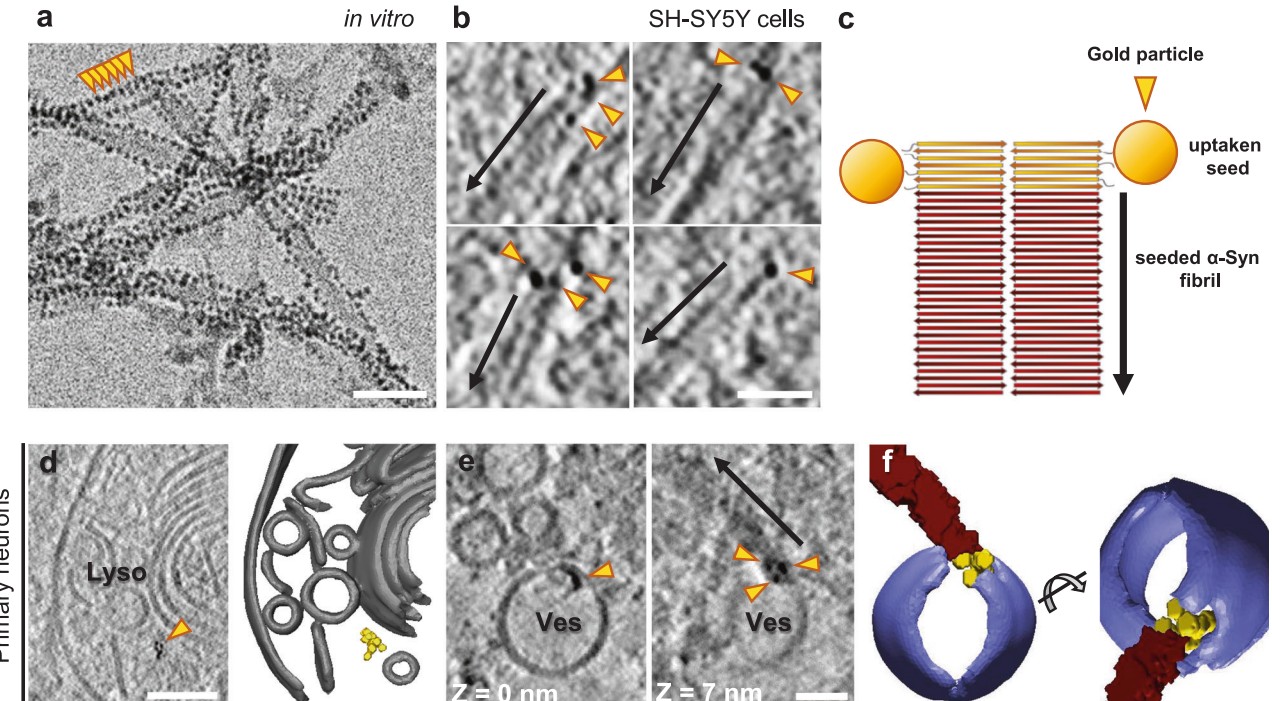

**Fig. 3 Seeding of α-Syn aggregates by gold-labeled PFFs. a** Cryo-electron microscopy image of PFFs labeled with 3-nm gold beads (orange arrowheads) via NHS esterification. Scale bar: 30 nm. Three biologically independent experiments were performed. The average distance between gold bead centers was 35 ± 4 Å (mean ± sd). Considering a typical distance between β strands of ~4.8 Å within α-Syn fibrils[33,34], this corresponds to labeling every seventh or eighth β-strand on the fibril. **b** Tomographic slices (thickness 1.8 nm) showing α-Syn fibrils seeded by gold-labeled PFFs within SH-SY5Y cells expressing GFP-α-Syn. Arrows mark the direction of fibril growth from the gold-labeled seed. Scale bar: 40 nm. Two biologically independent experiments were performed. **c** Schematic of the hypothetical molecular organization of α-Syn fibrils seeded by gold-labeled PFFs. **d** A tomographic slice (thickness 1.4 nm; left) and 3D rendering of the tomogram (right) showing gold-labeled PFFs within the lumen of a lysosome in a primary neuron expressing GFP-α-Syn. Lyso; lysosome. Lysosomal membranes (gray), gold particles labeling the PFF (yellow). Scale bar: 70 nm. **e** Tomographic slices (thickness 1.4 nm) at different Z heights showing gold-labeled PFFs found within the membrane of a vesicle (Ves) and seeding an α-Syn fibril (arrow) in a primary neuron expressing GFP-α-Syn. Scale bar: 30 nm. Two biologically independent experiments were performed. **f** 3D rendering of the tomogram depicted in **e** in two different orientations. Vesicle membrane (purple), α-Syn fibril (red), gold particles (yellow). Representative images are shown in **a**, **b**, **d**, **e**, and **f**.

results indicate that direct interactions between α-Syn fibrils and membranes are infrequent, and unlikely to induce substantial membrane clustering.

However, the previously suggested membrane clustering[20,21] could also be driven by α-Syn species smaller than fibrils, which cannot be readily detected by cryo-ET. For example, soluble α-Syn molecules can cluster vesicles at distances shorter than 15 nm in vitro[44]. To explore this possibility, we compared the shortest distances between all cellular membranes in tomograms of α-Syn inclusions and in untransduced, unseeded control neurons. This analysis revealed that close contacts (<20 nm) between membranes were similarly common within α-Syn inclusions as in control cells (Fig. 4c, Supplementary Fig. 7e, and Supplementary Table 1), arguing against α-Syn-mediated membrane clustering in inclusions.

Altogether, we show that neuronal α-Syn aggregates consist of both α-Syn fibrils and various cellular membranes. In agreement with a recent report[31], our findings suggest that the fibrils observed in pathological α-Syn inclusions[12–14,16] are indeed α-Syn fibrils. Intracellular α-Syn aggregation can be triggered by internalized small fibrils, suggesting that this mechanism is relevant to the spreading of aggregate pathology. However, α-Syn fibrils do not interact with membranes and do not seem to induce the type of membrane damage observed for other amyloids[22,45]. At the same time, α-Syn does not drive membrane clustering directly. Thus, it remains to be elucidated why membrane structures are enriched in α-Syn inclusions[12,16,31], in comparison with other neurotoxic protein aggregates[22–24]. An intriguing

possibility is that the abundance of vesicular organelles in α-Syn inclusions is caused by the impairment of the autophagic and endolysosomal machineries by α-Syn aggregation[46].

## Methods

**Plasmids**. Plasmids for the expression of recombinant α-Syn were: pT7-7 α-Syn (Addgene plasmid #36046 (ref. [47]); http://n2t.net/addgene:36046; RRID: Addgene_36046) and pT7-7 α-SynA53T (Addgene plasmid #105727 (ref. [48]); http://n2t.net/addgene:105727; RRID: Addgene_105727; gift from Hilal Lashuel).

Plasmid EGFP-α-SynA53T (Addgene plasmid #40823 (ref. [49]); http://n2t.net/addgene:40823; RRID: Addgene_40823) was used for expression in SH-SY5Y cells (gift from David Rubinsztein).

The following plasmids were used for viral transfections: pFhSynW2 (ref. [50]; GFP-SynA53T-Flag, Flag-GFP), FU3a (p62-tagRFP)[23], psPAX2 (a gift from Didier Trono; Addgene plasmid #12260; http://n2t.net/addgene:12260; RRID: Addgene_12260), and pVsVg[51]. pFhSynW2 and pVsVg were a gift of Dieter Edbauer.

pFhSynW2 GFP-synA53T-Flag was cloned by inserting the GFP-α-SynA53T sequence from plasmid EGFP-α-SynA53T between the XmaI and NheI restriction sites, using the primers described in Supplementary Table 2.

pFhSynW2 Flag-GFP was cloned by inserting the GFP sequence from the plasmid EGFP-α-SynA53T between the BamHI and EcoRI restriction sites, using the primers described in Supplementary Table 2.

**Antibodies**. The following primary antibodies were used: GFP (A10262, Thermo Fisher, 1:500; RRID: AB_2534023), K48-linked ubiquitin (05-1307, Millipore; 1:500; RRID: AB_1587578), MAP2 (NB300-213, Novus Biologicals; 1:500; RRID: AB_2138178), p62 (ab56416, Abcam; 1:200; RRID: AB_945626), phospho-α-Syn Ser129 (ab51253, Abcam; 1:500 for immunofluorescence, 1:2500 for western blot; RRID: AB_869973), α-Syn (610787, BD Biosciences; 1:1000; RRID: AB_398108), and p62 lck ligand (610832, BD Biosciences; 1:100; RRID: AB_398151).

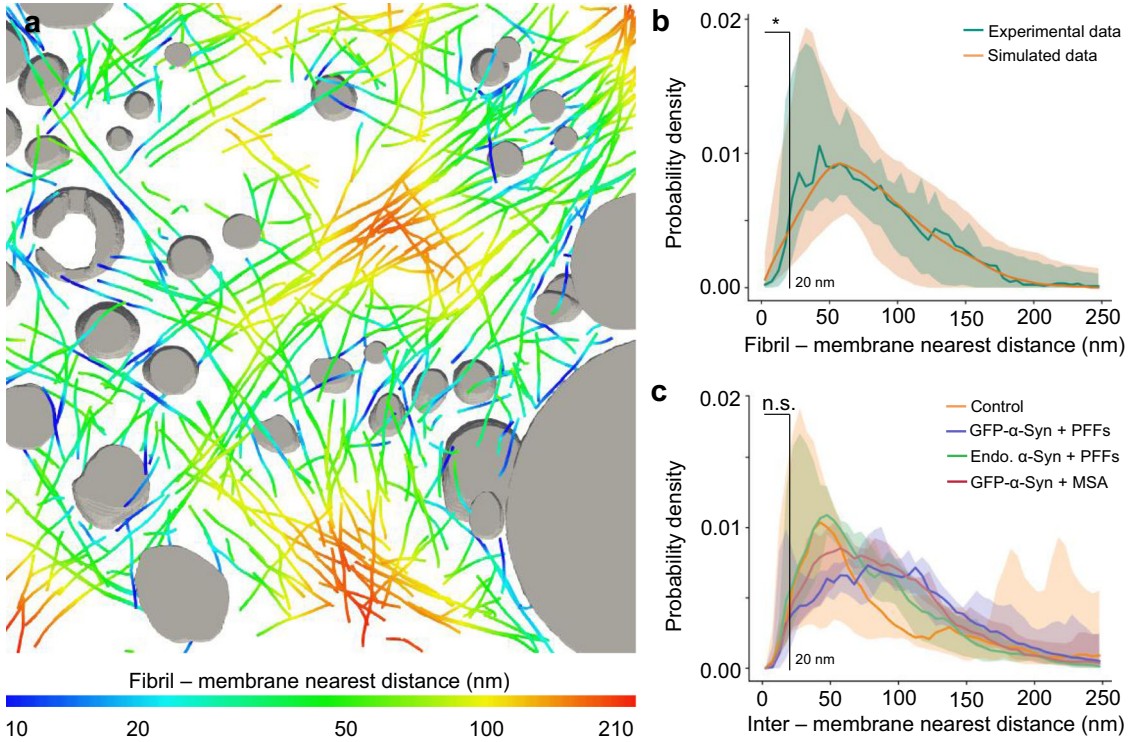

**Fig. 4 Quantification of fibril-membrane and inter-membrane distances within α-Syn inclusions. a** Visualization of fibril-membrane distances in the tomogram rendered in Fig. 1d. Organelles are shown in gray, and fibrils are color-coded according to their distance to the nearest organelle membrane. **b** Histogram of nearest distances between a fibril and a membrane in the pooled experimental data ($n = 15$ tomograms, including 6 of GFP-α-Syn + PFFs, 4 of endogenous α-Syn + PFFs, and 5 of GFP-α-Syn + MSA over three biologically independent experiments for all conditions), and in simulations shifting and rotating fibrils from their experimentally determined positions (200 simulations for each experimental tomogram). Solid lines represent the median of all tomograms. The shaded areas represent 5–95% confidence intervals. Fibril-membrane distances <20 nm are significantly more abundant in the simulated data ($p = 0.03$ by two-tailed Kolmogorov–Smirnov test). See also Supplementary Table 1. **c** Histogram of inter-membrane nearest distances for all organellar membranes in the tomograms. Inter-membrane distances <20 nm are not significantly different within α-Syn inclusions than in control untransduced and unseeded cells ($p = 0.4$ by two-tailed Kolmogorov–Smirnov test). Solid lines represent medians, shaded areas represent 5–95% confidence intervals. $n = 5$ (untransduced − PFFs), 6 (GFP-α-Syn + PFFs), 4 (endogenous α-Syn + PFFs), and 5 (GFP-α-Syn + MSA) tomograms analyzed over two (untransduced − PFFs) or three biologically independent experiments (GFP-α-Syn + PFFs, endogenous α-Syn + PFFs, and GFP-α-Syn + MSA). The number of tomograms and biologically independent cryo-ET experiments is listed in Supplementary Table 1. A representative image is shown in **a**. Source data for **b** and **c** are provided as a Source data file.

The following secondary antibodies were used: Alexa Fluor 488 AffiniPure Donkey anti-chicken (703-545-155, Jackson ImmunoResearch; 1:250), Alexa Fluor 647 AffiniPure Donkey anti-chicken (703-605-155, Jackson ImmunoResearch; 1:250), Cy3 AffiniPure Donkey anti-rabbit (711-165-152, Jackson ImmunoResearch; 1:250), Alexa Fluor 488 AffiniPure Donkey anti-mouse (715-545-150, Jackson ImmunoResearch; 1:250), Cy3-conjugated AffiniPure Goat anti-mouse IgG (115-165-003, Jackson ImmunoResearch; 1:1000), Cy3-conjugated AffiniPure Goat anti-rabbit (111-165-045, Dianova; 1:1000; RRID: AB_2338003), and HRP-conjugated goat anti-rabbit (A9169, Sigma; 1:5000; RRID: AB_258434).

**Recombinant α-Syn purification and fibril assembly.** Recombinant human WT and A53T α-Syn were purified as follows based on a published procedure[25] (see https://edmond.mpdl.mpg.de/imeji/collection/dBIbxxKvWaYMpyhI for a detailed protocol). BL21 (DE3) *Escherichia coli* (Agilent) were transformed with pT7-7 α-Syn or pT7-7 α-SynA53T. Protein expression was induced by 1 mM IPTG for 4 h. Bacteria were harvested and pellets were lysed in high salt (HS) buffer (750 mM NaCl, 50 mM Tris, pH 7.6, and 1 mM 2,2′,2″,2‴-(ethane-1,2-diyldinitrilo) tetraacetic acid (EDTA)). The lysate was sonicated for 5 min and boiled subsequently. The boiled suspension was centrifuged, the supernatant dialyzed in 50 mM NaCl, 10 mM Tris, and 1 mM EDTA, and purified by size-exclusion chromatography (Superdex 200). Fractions were collected and those containing α-Syn were combined. The combined fractions were applied onto an anion exchange column (MonoQ). α-Syn was purified by a gradient ranging from 50 mM to 1 M NaCl. Fractions containing α-Syn were combined and dialyzed in 150 mM KCl and 50 mM Tris, pH 7.6.

For fibril assembly, purified α-Syn monomers (5 mg/ml) were centrifuged at high speed ($100,000 \times g$) for 1 h. The supernatant was transferred into a new reaction tube, incubated with constant agitation (1000 r.p.m.) at 37 °C for 24 h. The resulting fibrils were diluted 1:20 in PBS and sonicated for 60 s (0.5 s on, 0.5 s off) using a Branson sonifier. The presence of α-Syn fibrils was confirmed by negative stain EM. Except for gold labeling experiments, cells were seeded using A53T α-Syn PFFs.

Labeling of fibrils with 3 nm monovalent gold beads (Nanopartz) via NHS ester coupling was performed, as described in the manufacturer's protocol. In brief, WT α-Syn PFFs were dialyzed in PBS and subsequently added to the gold beads. The reaction was facilitated by sonication (as above) and constant agitation at 30 °C for 30 min. Labeled PFFs and free gold beads were separated by sequential centrifugation and washing with 0.1% Tween20 and 1% PBS. Labeling of PFFs with gold beads was confirmed by negative stain and cryo-EM.

**Immunohistochemistry on MSA patient brain.** MSA patient brain tissue was obtained from Neurobiobank Munich (Germany). All autopsy cases of the Neurobiobank Munich are collected on the basis of an informed consent according to the guidelines of the ethics commission of the Ludwig-Maximilians-University Munich, Germany. The experiments performed in this paper were approved by the Max Planck Society's Ethics Council. The sample was from a male patient who died at the age of 54, 6 years after being diagnosed with a cerebellar type of MSA. Postmortem delay was ~30 h. Brain regions with abundant α-Syn inclusions were identified by postmortem histological examination.

For immunohistochemistry (IHC), mouse monoclonal antibodies against α-Syn and p62 lck ligand were used. Paraffin sections of human brain tissue were deparaffinized and rehydrated. Pretreatment (cooking in cell conditioning solution 1, pH 8 for 30 min for α-Syn IHC or for 56 min in case of p62 IHC), IHC, and counterstaining of nuclei with hematoxylin (Roche) and Bluing reagent (Roche) were performed with the Ventana Bench-Mark XT automated staining system (Ventana), using the UltraView Universal DAB Detection Kit (Roche). For α-Syn, IHC slides were additionally pretreated in 80% formic acid for 15 min after cooking. Slides were coverslipped with Entellan (Merck) mounting medium.

Images were recorded with a BX50 microscope (Olympus) using a 40× objective and cellSens 2.1 software (Olympus).

**Preparation of sarkosyl-insoluble fraction from MSA patient brain**. The sarkosyl-insoluble fraction was prepared as follows based on a published procedure[42]. Frozen tissue from the basilar part of the pons (1 cm$^3$) was homogenized in HS buffer (50 mM Tris-HCl pH 7.5, 750 mM NaCl, 10 mM NaF, and 5 mM EDTA) with protease and phosphatase inhibitors (Roche), and incubated on ice for 20 min. The homogenate was centrifuged at $100,000 \times g$ for 30 min. The resulting pellet was washed with HS buffer and then re-extracted sequentially with 1% Triton X-100 in HS buffer, 1% Triton X-100 in HS buffer and 30% sucrose, and 1% sarkosyl in HS buffer and finally PBS. The incubation with 1% sarkosyl in HS buffer was performed overnight at 4 °C. The final fraction was sonicated and the presence of α-Syn aggregates was confirmed by immunoblotting against phospho-α-Syn (Ser129) upon SDS–PAGE. Uncropped SDS–PAGE and immunoblots are shown in the Source data file.

**Cell culture**. To create a stable cell line expressing EGFP-α-SynA53T, SH-SY5Y cells (ACC209, DSMZ; RRID: CVCL_0019) were transfected using Lipofectamine 2000 (Thermo Fisher). Cells were cultured in in Dulbecco's modified Eagle's medium (DMEM, Biochrom) supplemented with 10% fetal bovine serum (FBS, GIBCO), 2 mM L-glutamine (GIBCO), and 1000 μg/ml geneticin for selection. Polyclonal cell lines were generated by fluorescence-activated cell sorting (FACS) with a BD FACS Aria III using FACSDiva 6.1.3 software. Upon selection, cells were cultured in medium supplemented with 200 μg/ml geneticin (Thermo Fisher) and penicillin/streptomycin (Thermo Fisher).

Cells were seeded as described[25] using 300 nM (monomer) of α-SynA53T PFFs or gold-conjugated WT α-Syn PFFs. In brief, sonicated PFFs were diluted in a mixture of 50 μl of Optimem (Biochrom) and 3 μl of Lipofectamine 2000. Subsequently, the suspension was added to 1 ml of cell culture medium.

**Lentivirus packaging**. HEK293T cells (632180, Lenti-X 293T cell line, Takara; RRID: CVCL_0063) for lentiviral packaging were expanded to 70–85% confluency in DMEM Glutamax (+4.5 g/l D-glucose, -pyruvate) supplemented with 10% FBS (Sigma), 1% G418 (Gibco), 1% NEAA (Thermo Fisher), and 1% Hepes (Biomol). Only low passage cells were used. For lentiviral production, a three-layered 525 cm$^2$ flask (Falcon) was seeded and cells were henceforth cultured in medium without G418. On the following day, cells were transfected with the expression plasmid pFhSynW2 (GFP-SynA53T-Flag, Flag-GFP) or FU3a (p62-tagRFP), and the packaging plasmids psPAX2 and pVsVg using TransIT-Lenti transfection reagent (Mirus). The transfection mix was incubated for 20 min at room temperature (RT) and cell medium was exchanged. A total of 10 ml of transfection mix were added to the flask and incubated overnight. The medium was exchanged on the next day. After 48–52 h, culture medium containing the viral particles was collected and centrifuged for 10 min at $1200 \times g$. The supernatant was filtered through 0.45 μm pore size filters using 50 ml syringes, and Lenti-X concentrator (Takara) was added. After an overnight incubation at 4 °C, samples were centrifuged at $1500 \times g$ for 45 min at 4 °C, the supernatant was removed, and the virus pellet was resuspended in 600 μl TBS-5 buffer (50 mM Tris-HCl, pH 7.8, 130 mM NaCl, 10 mM KCl, and 5 mM MgCl$_2$). After aliquoting, viruses were stored at −80 °C.

**Primary neurons**. Primary cortical neurons were prepared from E15.5 CD-1 wild-type mouse embryos of both sexes (breeding line MpiCrlIcr:CD-1). Mice were housed in a specific pathogen-free facility at 22 ± 1,5 °C, 55 ± 5% humidity, 14-h light/10-h dark cycle. All experiments involving mice were performed in accordance with the relevant guidelines and regulations of the Government of Upper Bavaria (Germany). Pregnant females were sacrificed by cervical dislocation, the uterus was removed from the abdominal cavity and placed into a 10 cm sterile Petri dish on ice containing dissection medium, consisting of Hank's balanced salt solution supplemented with 0.01 M HEPES, 0.01 M MgSO$_4$, and 1% penicillin/streptomycin. Embryos of both sexes were chosen randomly. Each embryo was isolated, heads were quickly cut, and brains were removed from the skull and immersed in ice-cold dissection medium. Cortical hemispheres were dissected and meninges were removed under a stereo-microscope. Cortical tissue from typically six to seven embryos was transferred to a 15 ml sterile tube, and digested with 0.25% trypsin containing 1 mM EDTA and 15 μl 0.1% DNASe I for 20 min at 37 °C. The enzymatic digestion was stopped by removing the supernatant and washing the tissue twice with Neurobasal medium (Invitrogen) containing 5% FBS. The tissue was resuspended in 2 ml medium and triturated to achieve a single-cell suspension. Cells were spun at $130 \times g$, the supernatant was removed, and the cell pellet was resuspended in Neurobasal medium with 2% B27 (Invitrogen), 1% L-glutamine (Invitrogen), and 1% penicillin/streptomycin (Invitrogen). For immunostaining, neurons were cultured in 24-well plates on 13 mm coverslips coated with 1 mg/ml poly-D-lysine (Sigma) and 1 μg/ml laminin (Thermo Fisher Scientific; 100,000 neurons per well). For thiazolyl blue tetrazolium bromide (MTT) assay, neurons were cultured in 96-well plates coated in the same way (19,000 neurons per well). For Cryo-ET, EM grids were placed in 24-well plates and coated as above (120,000 neurons per well). For lentiviral transduction at DIV 10, viruses were thawed and immediately added to freshly prepared neuronal culture medium.

Neurons in 24-well plates received 1 μl of virus/well, while neurons in 96-well plates received 0.15 μl of virus/well. A fifth of the medium from cultured neurons was removed and the equivalent volume of virus-containing medium was added. Three days after transduction, 2 μg/ml of seeds (α-SynA53T PFFs, gold-conjugated WT α-Syn PFFs, or MSA-derived aggregates) were added to the neuronal culture medium.

**MTT viability assay**. Viability of transduced neurons was determined using MTT (Sigma-Aldrich). Cell medium was exchanged for 100 μl of fresh medium, followed by addition of 20 μl of 5 mg/ml MTT in PBS and incubation for 2–4 h at 37 °C, 5% CO$_2$. Subsequently, 100 μl solubilizer solution (10% SDS and 45% dimethylformamide in water, pH 4.5) was added, and on the following day absorbance was measured at 570 nm. Three biologically independent experiments were performed for each condition, and absorbance values were averaged for each experiment. Viability values of neurons seeded with α-Syn aggregates were normalized to those of neurons that received PBS only.

**Immunofluorescence**. Immunofluorescence stainings on SH-SY5Y cells were performed 24 h after seeding. Cells were fixed for 10 min with 4% paraformaldehyde (PFA) in PBS and subsequently incubated for 5 min in permeabilization solution (0.1% Triton X-100 in PBS) at RT. After blocking with 5% milk in permeabilization solution, primary antibodies were diluted in blocking solution and incubated with the cells overnight at 4 °C. Secondary antibodies were incubated with the cells in blocking solution for 3 h at RT. The coverslips were subsequently incubated with 500 nM DAPI for 10 min and then mounted on glass slides. Images were taken using a CorrSight microscope (Thermo Fisher) in spinning disc mode with a 63× oil immersion objective.

Primary neurons were fixed with 4% PFA in PBS for 20 min; remaining free groups of PFA were blocked with 50 mM ammonium chloride in PBS for 10 min at RT. Cells were rinsed once with PBS and permeabilized with 0.25% Triton X-100 in PBS for 5 min. After washing with PBS, blocking solution consisting of 2% BSA (Roth) and 4% donkey serum (Jackson ImmunoResearch) in PBS was added for 30 min at RT. Coverslips were transferred to a light protected humid chamber and incubated with primary antibodies diluted in blocking solution for 1 h. Cells were washed with PBS and incubated with secondary antibodies diluted 1:250 in blocking solution, with 0.5 μg/ml DAPI added to stain the nuclei. Coverslips were mounted on Menzer glass slides using Prolong Glass fluorescence mounting medium. Confocal images were obtained at a SP8 confocal microscope (Leica) using 40× or 63× oil immersion objectives and Las X 3.5.7.23225 software (Leica). Neurons containing aggregates in the soma were manually quantified using the Cell Counter plugin of ImageJ 2.0.0 (ref. [52]; RRID: SCR_003070).

**Negative stain EM**. For negative stain analysis, continuous carbon Quantifoil grids (Cu 200 mesh, QuantifoilMicro Tools) were glow discharged using a plasma cleaner (PDC-3XG, Harrick) for 30 s. Grids were incubated for 1 min with PFFs, blotted and subsequently washed two times with water for 10 s. The blotted grids were stained with 0.5% uranyl acetate solution, dried, and imaged in a Polara cryo-electron microscope (Thermo Fisher) operated at 300 kV, using a pixel size of 2.35 or 3.44 Å.

**Cryo-ET sample preparation**. Quantifoil grids (R1/4 or 1.2/20, Au mesh grid with SiO$_2$ film, QuantifoilMicro Tools) were glow discharged using a plasma cleaner (PDC-3XG, Harrick) for 30 s. Cells were plated on the grids as described above. SH-SY5Y cells were seeded with α-Syn aggregates 24 h after plating and plunge frozen after another 24 h. Neurons were transduced on DIV 10, seeded with α-Syn aggregates on DIV 13, and plunge frozen on DIV 20. Plunge freezing was performed on a Vitrobot (Thermo Fisher) with the following settings: temperature, 37 °C; humidity, 80%; blot force, 10; and blot time, 10 s. The grids were blotted from the back and the front using Whatman filter paper and plunged into a liquid ethane/propane mixture. Subsequently the vitrified samples were transferred into cryo-EM boxes and stored in liquid nitrogen.

**Correlative cryo-light microscopy and cryo-FIB milling**. Grids were mounted onto autogrid sample carriers (Thermo Fisher) that contain cutout regions to facilitate shallow-angle FIB milling. Subsequently, grids were transferred into the stage of a CorrSight cryo-light microscope (Thermo Fisher) cooled at liquid nitrogen temperature. Overview images of the grids were acquired using a 20× lens (air, N.A., 0.8). Cells containing fluorescence signal of interest (GFP-α-Syn or p62-RFP) were mapped using MAPS 2.1 software (Thermo Fisher; RRID: SCR_018738).

The samples were transferred into a Scios or Quanta dual beam cryo-FIB/scanning electron microscopes (SEM; Thermo Fisher). To avoid charging of the samples, a layer of inorganic platinum was deposited on the grids. That was followed by the deposition of organometallic platinum using an in situ gas injection system (working distance—10 mm, heating—27 °C, time—8 s) to avoid damaging the cells by out-of-focus gallium ions. Subsequently, 2D correlation was performed using MAPS and the three-point alignment method between the fluorescence and the SEM image, as described[23].

For FIB milling, the grid was tilted to 18° and gallium ions at 30 kV were used to remove excess material from above and below the region of interest. Rough milling was conducted at a current of 0.5 nA and fine milling at a current of 50 pA, resulting in 100–200 nm thick lamellas (see https://edmond.mpdl.mpg.de/imeji/collection/dBIbxxKvWaYMpyhI for a detailed protocol of these procedures).

**Cryo-ET data collection and reconstruction.** The lamellas were transferred into a Titan Krios cryo-electron microscope (Thermo Fisher) operated at 300 kV, and subsequently loaded onto a compustage cooled to liquid nitrogen temperatures. Lamellas were oriented perpendicular to the tilt axis. Images were collected using a $4\,k \times 4\,k$ K2 Summit or K3 (Gatan) direct detector cameras operated in dose fractionation mode (0.2 s, 0.15 e$^{-}$/Å$^2$). A BioQuantum (Gatan) post column energy filter was used with a slit width of 20 eV. Tilt series were recorded using SerialEM 3.7.0 (ref. [53]; RRID: SCR_017293) at pixel size of 3.38, 3.52, or 4.39 Å. Tilt series were recorded dose symmetrically[54] from $-50°$ to $+60°$ with an angular increment of 2°, resulting in a total dose of 100–130 e$^{-}$/Å$^2$ per tilt series. Frames were aligned using Motioncor2 1.2.1 (ref. [55]). Final tilt series were aligned using fiducial-less patch tracking, and tomograms were reconstructed by using back projection in IMOD 4.9.0 (ref. [56]; RRID: SCR_003297). Contrast was enhanced by filtering the tomograms using tom_deconv (https://github.com/dtegunov/tom_deconv) within MATLAB 2017a (MathWorks).

**Tomogram segmentation.** The membranes of the tomograms were segmented using the automatic membrane tracing package TomoSegMemTV[57]. The results were refined manually in Amira 6.2 (FEI Visualization Science Group; RRID: SCR_014305). The lumen of organelles was filled manually based on the membrane segmentations. For tracing of α-Syn fibrils, the XTracing module[58] of Amira was used. For that the tomograms were first denoised with a nonlocal means filter, and subsequently searched with a cylindrical template of 10 nm diameter and 80 nm length. Based on the cross-correlation fields, thresholds producing an optimal balance of true positives and negatives were applied. Filaments were subsequently traced with a search cone of 50 nm length and an angle of 37°. The direction coefficient was 0.3 and the minimum filament length was set to 100 nm. Selected filaments were inspected visually. Supplementary Movie 1 was created using Amira.

**Fibril diameter and density of gold labeling.** The diameter of endogenous α-Syn fibrils was measured using IMOD. 40 different positions were measured along different fibrils in three tomograms. For gold-labeled PFFs, the distance between the centers of gold particles was also measured with IMOD at 20 different positions. To estimate the labeling density, the average distance between gold particles was divided by 4.8 Å, approximately corresponding to the spacing between β-strands measured in α-Syn fibrils[33,34].

**Cytosolic fibril density.** The density of fibrils within the inclusion was calculated as the fraction of cytosolic volume occupied by fibrils. Cellular volume was calculated multiplying the X and Y dimensions of the tomogram by the thickness of the lamella along the Z direction. To calculate cytoplasmic volume, the lumina of organelles were subtracted from the tomogram volume. Fibril volume was calculated approximating fibrils by cylinders with radius of 5 nm and the length calculated by filament tracing. Calculations were performed in Origin 2019b (RRID: SCR_014212).

**Fibril persistence length.** The persistence length ($L_p$) measures the stiffness of polymers as the average distance for which a fibril is not bent. It was calculated using an in-house script, as previously described[22] executed in MATLAB. Briefly, $L_p$ is calculated as the expectation value of cos θ, where θ is defined as the angle between two tangents to the fibril at positions 0 and l (ref. [59]):

$$\langle \cos(\theta_0 - \theta_l) \rangle = e^{-(l/L_p)} \tag{1}$$

The Young's modulus ($E$) can be calculated from $L_p$ as:

$$E = \frac{L_p k_B T}{I} \tag{2}$$

where $k_B$ is the Boltzmann constant ($1.38 \times 10^{-23}$ m$^2$ kg s$^{-2}$ K$^{-1}$), $T$ is the absolute temperature (here 295 K), and $I$ is the momentum of inertia. Approximating the fibril by a solid rod, $I$ can be calculated from its radius $r$ as:

$$I = \frac{\pi r^4}{4} \tag{3}$$

Here, we used $r = 5$ nm.

Confidence intervals were calculated by fitting the data points to a linear polynomial using linear least squares in the MATLAB function "fit".

**Fibril-membrane distance.** Fibril-membrane nearest distance was defined for each point on the fibril as the minimum Euclidean distance to another point on a membrane. The algorithm computing fibril-membrane nearest distances is described in the Supplementary Methods.

**Inter-membrane distance.** Each segmented lumen was labeled differently to identify different organelles. The inter-membrane nearest distance for a point on a membrane was defined as the minimum Euclidean distance to another point on a membrane associated to a different lumen. The algorithm for computing inter-membrane nearest distances is described in the Supplementary Methods.

**Statistical analysis.** For the quantification of the percentage of neurons with aggregates using light microscopy (Supplementary Fig. 1f), $n = 4$ (GFP-α-Syn + PFFs) and 3 (endogenous α-Syn + PFFs) biologically independent experiments were performed, and a total of 100–500 neurons per condition and per experiment were counted. Statistical analysis was carried out by two-tailed unpaired $t$ test with Welch's correction in Prism 6 (GraphPad; RRID: SCR_002798).

For the quantification of neuronal viability using the MTT assay (Supplementary Fig. 1g), $n = 3$ biologically independent experiments were performed for all conditions. Untransduced and unseeded control cells were used as reference. Statistical analysis was carried out by two-way ANOVA and Dunnett's multiple comparison test in Prism 6.

The number of tomograms and fibrils, as well as the total membrane area analyzed for each condition are shown in Supplementary Table 1.

Statistical analysis of cytosolic fibril density (Fig. 2e) was carried out by one-way ANOVA in Origin (RRID:SCR_014212). Confidence intervals for fibril-membrane (Fig. 4b) and inter-membrane (Fig. 4c) distances were calculated as the 5–95 percentiles from the curves of each individual tomogram. The differences between the curves within 20 nm were statistically analyzed by Kolmogorov–Smirnov test. Additional information on statistical analyses can be found in the Source data file.

**Reporting summary.** Further information on research design is available in the Nature Research Reporting Summary linked to this article.

## Data availability
The data supporting the findings of this manuscript are available from the corresponding authors upon reasonable request. A reporting summary for this Article is available as a Supplementary Information file. The individual values for the average graphs shown in Figs. 2d and 4b, c, and Supplementary Fig. 4 are available at the Edmond repository: https://edmond.mpdl.mpg.de/imeji/collection/rnVkI2lwG8loNXOi. The tomograms shown in Figs. 1 and 2 are available in EMPIAR through accession codes EMD-11401 (Fig. 1a), EMD-11417 (Fig. 1e), and EMD-11416 (Fig. 2a). Source data are provided with this paper. Protocols for recombinant α-Syn purification and cryo-correlative microscopy are available at the Edmond repository: https://edmond.mpdl.mpg.de/imeji/collection/dBIbxxKvWaYMpyhI. All other data are available from the corresponding authors upon reasonable request.

## Code availability
The tomogram deconvolution filter is available at: https://github.com/dtegunov/tom_deconv. The script[60] for the calculation of $L_p$ is available at: https://github.com/FJBauerlein/Huntington. The scripts[61] for fibril-membrane and inter-membrane distance calculations were performed within the PySeg software[62] and are available at: https://github.com/anmartinezs/pyseg_system/tree/master/code/pyorg/scripts/filaments.

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

## Acknowledgements

We thank Philipp Erdmann, Günter Pfeifer, Jürgen Plitzko, and Miroslava Schaffer for electron microscopy support, and Ana Jungclaus and Nadine Wischnewski for wet lab support. We thank Dieter Edbauer, Hilal Lashuel, David Rubinsztein, and Didier Trono for sharing plasmids. We thank Konstanze Winklhofer and Joerg Tazelt for sharing the SH-SY5Y cell line and helpful discussions. We are also grateful to Sophie Keeling for help in plasmid cloning, Javier Collado for help with sample preparation, Jonathan Schneider and William Wan for help with image processing, Patrick Auer for help in aggregate

quantification, as well as Itika Saha, Eri Sakata, and Patricia Yuste-Checa for helpful discussions. Finally, we thank the anonymous MSA patient and his family for the donation of brain tissue to the Neurobiobank Munich. Fluorescence-activated cell sorting was carried out at the Imaging Facility of the Max Planck Institute of Biochemistry. V.A.T. was supported by the Graduate School of Quantitative Biosciences Munich. V.A.T., I.R.-T., A.M.-S., F.J.B., Q.G., W.B., I.D., M.S.H., F.U.H., and R.F.-B. have received funding from the European Commission (FP7 GA ERC-2012-SyG_318987-ToPAG). I.D. acknowledges financial support from the Horst Kübler-Stiftung. V.A.T., T.A., M.S.H., F.U.H., and R.F.-B. acknowledge funding from the Deutsche Forschungsgemeinschaft (DFG, German Research Foundation) through Germany's Excellence Strategy—EXC 2067/1—390729940 (R.F.-B.) and EXC 2145 – 390857198 (V.A.T., T.A., M.S.H. and F.U.H). F.U.H. and R.F.-B. were funded by the joint efforts of The Michael J. Fox Foundation for Parkinson's Research (MJFF) and the Aligning Science Across Parkinson's (ASAP) initiative. MJFF administers the grant ASAP-000282 on behalf of ASAP and itself. For the purpose of open access, the authors have applied a CC-BY public copyright license to the Author Accepted Manuscript version arising from this submission.

## Author contributions

V.A.T. performed biochemical and electron microscopy experiments, immuno-fluorescence imaging of SH-SY5Y cells, and contributed to computational data analysis. I.R.-T. produced lentivirus and neuronal cultures, and performed viability assays and immunofluorescence imaging of neurons. A.M.S. developed software procedures for data analysis. F.B. and Q.G. contributed to data analysis. T.A. collected the autopsy case, characterized it neuropathologically, and performed immunohistochemistry. V.A.T., I.R.-T., W.B., I.D., M.S.H., F.U.H., and R.F.-B. planned research. I.D. supervised neuronal culture experiments. M.S.H. and F.U.H. supervised biochemical experiments. R.F.-B. supervised electron microscopy experiments and data analysis. R.F.-B. wrote the manuscript with contributions from all authors.

## Funding

## Competing interests

The authors declare no competing interests.
