## [Peer Review File · Nature Communications]

Reviewer #1 (Remarks to the Author):

The authors used cryo-electron tomography to image seeded alpha-synuclein filaments in primary neurons (untransfected or transfected) and SH-SY5Y cells (transfected). The seeds were either sonicated preformed fibrils (PFFs) assembled from human recombinant wild-type or mutant A53T alpha-synuclein, gold-labelled or not, or prepared from the pons of a case of neuropathologically confirmed multiple system atrophy (MSA) using sarkosyl extraction and sonication.

The tomography part of this work is of high quality and wide interest. However, the description of seed production lacks in detail (see point 3 of major comments). Moreover, discussion of some of the background and implications is inadequate and needs changing.

Major comments

1. The authors repeatedly mention the paper by Shahmoradian et al. (2019), which questioned the relevance of alpha-synuclein filaments for Lewy pathology. The present work cannot say anything definitive about this controversy, because it is unclear how seeded aggregation using PFFs relates to Lewy pathology. As the authors mention, the structures of filaments made from recombinant protein may differ from those found in human brain. The other seeds were prepared from MSA brain. But MSA is not characterized by Lewy pathology. So, although this work deals with seeded aggregation of alpha-synuclein, it cannot say anything specifically about Lewy bodies. It is incorrect to call the assemblies observed here 'Lewy body-like'. This must be changed. This also applies to the claim that, based on the current work, the filaments seen by Shahmordian et al. were alpha-synuclein filaments. The paper by Mahul-Mellier, which is directly relevant for the mechanisms underlying Lewy body formation, must be discussed and cited (PNAS 117, 4971-4982, 2020). Perhaps it is all only question of time downstream of filament assembly. The work by Shahmoradian et al. may or may not be relevant for human diseases. One way to link what is going on in human diseases with what is described here would be to use labels that are positive in both. This may go beyond the scope of this manuscript, but the possibility should be mentioned. Such labels could be thioflavins, luminescent conjugated oligothiophenes or silver stains.

2. The work showing polarised growth of alpha-synuclein filaments is interesting. The authors show that upon internalisation of gold-labelled seeds made of wild-type alpha-synuclein PFFs, seeded GFP-alpha-synuclein filaments were only decorated at one end by gold particles. But the experiments used recombinant assembled proteins and the reference (Guerrero-Ferreira et al., 2019) refers to the cryo-EM structures of filaments assembled from recombinant (1-121) alpha-synuclein. As mentioned above, the structures of recombinant alpha-synuclein filaments are probably different from those found in human brain. The authors ignore the evidence (from over 20 years ago) suggesting polarised growth of alpha-synuclein filaments in human brain (dementia with Lewy bodies, MSA, Parkinson's disease). References 5 and 6, as well as Crowther et al. Neuroscience Letters 292, 128-130 (2000) must be cited in this respect. The authors must discuss how they see polarised growth of alpha-synuclein filaments in MSA, taking into account the cryo-EM structures of alpha-synuclein filaments from MSA described in reference 26.

3. The authors say that nucleation-relevant seeds consist of oligomeric alpha-synuclein. They should explain this statement. Is a major point of seeded aggregation not that it is nucleation-independent (no lag phase)? In the Methods it says that seeds were sonicated. The authors must show negative stain pictures of their seeds after sonication and describe how sonication was carried out. This is a

crucial piece of information. How can they exclude that what they call oligomers were in fact short filaments? If they cannot do so, they must mention the possibility that at least some seeds were short filaments of alpha-synuclein. The paper by Pieri et al. (Scientific Reports 6: 24526, 2016) must be discussed and cited. Can tomography distinguish between large oligomers and small filaments?

4. This study depends on seeded aggregate formation. It is important to point out that there is no evidence to suggest that the high-resolution structures of seeds and seeded aggregates are the same, which is a hidden assumption behind much of the discussion.

Minor comments

1. Duffy and Tennyson (JNEN 24, 398-414, 1965) described Lewy body filaments before Roy and Wolman.

2. Cross-reaction of alpha-synuclein antibodies with neurofilaments is not a general problem. Reference 12 and its discussion must be removed.

3. The work of Araki et al. (PNAS 116, 17963-17969, 2019) must be discussed and cited. It looks at human disease and disagrees with the conclusions of Shahmoradian et al.

Reviewer #2 (Remarks to the Author):

Review for Trinkhaus et al

Trinkhaus et al present a methodologically sound, well-controlled study showing a higher resolution picture of the cellular and molecular organisation of Lewy bodies, neuronal inclusions common to several neurodegenerative diseases, including Parkinson's Disease and multiple systems atrophy. Previous studies - most notably a room temperature correlative light and electron microscope and tomography study - have shown that alpha-synuclein-positive Lewy bodies consist of crowded environments of membranes of disfigured organelles and vesicles, as well as unidentified filamentous structures. The latter have been proposed to consist of alpha-synuclein, but their identity has not been unequivocally confirmed due to lower resolution of the employed imaging method.

There is very little to fault this study, as it is fairly coherent and well-designed. Only the following minor points -

- figure 3a – beautiful images. How does the labelling relate / agree with the atomic structure of Syn reported by Henning Stahlberg?

- “Our measurements are also consistent with single-particle studies reporting a higher twist, indicative of higher flexibility, for MSA-derived fibrils compared to recombinant”

Could you provide numbers for twists, diameters, maybe even an image comparison with the downloaded EMDB structure?

- Perhaps use of red and green in figures could be prevented? I suggest to change to cyan and magenta throughout the manuscript.

- L286: The caption reads '3D rendering of a)', but should be '3D rendering of tomogram depicted in a)', as the description for a) reads 'tomographic slice', which cannot be 3D-rendered.

- L304: 'Cytosolic fibril density': It is not directly obvious what is meant by this term as it is only explained in the methods section. Perhaps this could be explained in the main text?

- L556: 'cameras' -> 'camera'

We would like to thank the reviewers for their comments, which we found very helpful in improving the manuscript.

Reviewer #1 (Remarks to the Author):

The authors used cryo-electron tomography to image seeded alpha-synuclein filaments in primary neurons (untransfected or transfected) and SH-SY5Y cells (transfected). The seeds were either sonicated preformed fibrils (PFFs) assembled from human recombinant wild-type or mutant A53T alpha-synuclein, gold-labelled or not, or prepared from the ponds of a case of neuropathologically confirmed multiple system atrophy (MSA) using sarkosyl extraction and sonication.

The tomography part of this work is of high quality and wide interest. However, the description of seed production lacks in detail (see point 3 of major comments). Moreover, discussion of some of the background and implications is inadequate and needs changing.

Major comments

1. The authors repeatedly mention the paper by Shahmoradian et al. (2019), which questioned the relevance of alpha-synuclein filaments for Lewy pathology. The present work cannot say anything definitive about this controversy, because it is unclear how seeded aggregation using PFFs relates to Lewy pathology. As the authors mention, the structures of filaments made from recombinant protein may differ from those found in human brain. The other seeds were prepared from MSA brain. But MSA is not characterized by Lewy pathology. So, although this work deals with seeded aggregation of alpha-synuclein, it cannot say anything specifically about Lewy bodies. It is incorrect to call the assemblies observed here 'Lewy body-like'. This must be changed. This also applies to the claim that, based on the current work, the filaments seen by Shahmoradian et al. were alpha-synuclein filaments. The paper by Mahul-Mellier, which is directly relevant for the mechanisms underlying Lewy body formation, must be discussed and cited (PNAS 117, 4971-4982, 2020). Perhaps it is all only question of time downstream of filament assembly. The work by Shahmoradian et al. may or may not be relevant for human diseases. One way to link what is going on in human diseases with what is described here would be to use labels that are positive in both. This may go beyond the scope of this manuscript, but the possibility should be mentioned. Such labels could be thioflavins, luminescent conjugated oligothiophenes or silver stains.

We no longer refer to the α -Syn inclusions observed in our cellular system as "Lewy body-like". We have also rephrased the statements on the disease-relevance of our findings and refer to Mahul-Mellier et al. (ref 31), e.g. (lines 198-200): "Altogether, we show that neuronal α -Syn aggregates consist of both α -Syn fibrils and various cellular membranes. In agreement with a recent report³¹, our findings suggest that the fibrils observed in pathological α -Syn inclusions^{12-14,16} are indeed α -Syn fibrils."

Moreover, we have included additional references illustrating the similarities between seeded aggregation with PFFs and Lewy bodies, including thioflavin staining (ref 26, lines 81-83): "We performed cryo-ET on neuronal α -Syn aggregates using a well-established seeding paradigm that recapitulates inter-neuronal spreading and key neuropathological features of Lewy bodies, including the ability to bind amyloid dyes^{1,25-27}."

2. The work showing polarised growth of alpha-synuclein filaments is interesting. The authors show that upon internalisation of gold-labelled seeds made of wild-type alpha-synuclein PFFs, seeded GFP-alpha-synuclein filaments were only decorated at one end by gold particles. But the experiments used recombinant assembled proteins and the reference (Guerrero-Ferreira et al., 2019) refers to the cryo-EM structures of filaments assembled from recombinant (1-121) alpha-synuclein. As mentioned above, the structures of recombinant alpha-synuclein filaments are probably different from those found in human brain. The authors ignore the evidence (from over 20 years ago) suggesting polarised growth of alpha-synuclein filaments in human brain (dementia with Lewy bodies, MSA, Parkinson's disease). References 5 and 6, as well as Crowther et al. *Neuroscience Letters* 292, 128-130 (2000) must be cited in this respect. The authors must discuss how they see polarised growth of alpha-synuclein filaments in MSA, taking into account the cryo-EM structures of alpha-synuclein filaments from MSA described in reference 26.

The suggested references and discussion regarding the polarized growth of α -Syn fibrils are now included (lines 153-158): “Interestingly, cryo-ET analysis of inclusions seeded by gold-labeled PFFs showed GFP- α -Syn fibrils with one end decorated by 3-10 gold particles (Fig. 3b, c), indicating that exogenous seeds trigger the fibrillation of cellular α -Syn in a polarized manner, consistent with the polar structures of recombinant α -Syn fibrils³⁴. Although patient-derived α -Syn fibrils are polar as well^{7,8,33,40}, it remains to be established whether disease-related seeds also trigger unidirectional fibril growth in cells.”

3. The authors say that nucleation-relevant seeds consist of oligomeric alpha-synuclein. They should explain this statement. Is a major point of seeded aggregation not that it is nucleation-independent (no lag phase)? In the Methods it says that seeds were sonicated. The authors must show negative stain pictures of their seeds after sonication and describe how sonication was carried out. This is a crucial piece of information. How can they exclude that what they call oligomers were in fact short filaments? If they cannot do so, they must mention the possibility that at least some seeds were short filaments of alpha-synuclein. The paper by Pieri et al. (*Scientific Reports* 6: 24526, 2016) must be discussed and cited. Can tomography distinguish between large oligomers and small filaments?

We have now modified the corresponding sections, avoiding the term “nucleation”.

Given that our seeds were produced by sonication of mature fibrils, we agree that it is more likely that the small seeding-competent species correspond to small fibrils, rather than oligomers. This has now been corrected throughout. This conclusion is consistent with the findings of Pieri et al., which we now cite (ref 41, lines 158-160): “Our data also show that, in our experimental conditions, small α -Syn fibrils are the most seeding-competent species, in agreement with previous results⁴¹.”

Additional information on the sonication procedure is now included in the Methods (lines 332-333, 338-339), and negative stain images are shown in Supplementary Fig. 1b.

4. This study depends on seeded aggregate formation. It is important to point out that there is no evidence to suggest that the high-resolution structures of seeds and seeded aggregates are the

same, which is a hidden assumption behind much of the discussion.

The following sentence has been added to clarify this point (lines 165-168): “Although α -Syn strains can be transmitted between cells *in vitro* and *in vivo*³, the cellular environment may also modify the strain characteristics⁴². Thus, higher resolution data are needed to elucidate to what extent the structure of the seed is templated in the aggregates seeded within cells.”

Minor comments

1. Duffy and Tennyson (JNEN 24, 398-414, 1965) described Lewy body filaments before Roy and Wolman.

We thank the Reviewer for pointing us to this study, which is now also cited in our manuscript (ref 11, lines 58-59): “Early electron microscopy (EM) studies suggested that the Lewy bodies⁹⁻¹² and glial cytoplasmic inclusions^{13,14},”

2. Cross-reaction of alpha-synuclein antibodies with neurofilaments is not a general problem. Reference 12 and its discussion must be removed.

Done.

3. The work of Araki et al. (PNAS 116, 17963-17969, 2019) must be discussed and cited. It looks at human disease and disagrees with the conclusions of Shahmoradian et al.

We now cite Araki et al. in the Introduction (ref 15, lines 62-63): “Intriguingly, X-ray diffraction measurements confirmed the presence of amyloid fibrils only in some Lewy bodies¹⁵”

Reviewer #2 (Remarks to the Author):

Review for Trinkhaus et al

Trinkhaus et al present a methodologically sound, well-controlled study showing a higher resolution picture of the cellular and molecular organisation of Lewy bodies, neuronal inclusions common to several neurodegenerative diseases, including Parkinson's Disease and multiple systems atrophy. Previous studies - most notably a room temperature correlative light and electron microscope and tomography study - have shown that alpha-synuclein-positive Lewy bodies consist of crowded environments of membranes of disfigured organelles and vesicles, as well as unidentified filamentous structures. The latter have been proposed to consist of alpha-synuclein, but their identity has not been unequivocally confirmed due to lower resolution of the employed imaging method.

There is very little to fault this study, as it is fairly coherent and well-designed. Only the following minor points –

We thank the Reviewer for the overall positive comments.

- figure 3a – beautiful images. How does the labelling relate / agree with the atomic structure of Syn reported by Henning Stahlberg?

We measured an average distance between gold bead centers of 3.5 ± 0.4 nm (mean \pm sd) along α -Syn fibrils. Given that the typical distance between beta strands is ~ 4.8 Å across various high-resolution structures of α -Syn fibrils (PMID 32112991, 32461689), we estimate that the gold beads label approximately every 7-8 beta strands. This is now mentioned in the legend of Fig. 3 (lines 247-249). The cartoon in Fig. 3c has also been re-scaled accordingly.

- “Our measurements are also consistent with single-particle studies reporting a higher twist, indicative of higher flexibility, for MSA-derived fibrils compared to recombinant”
Could you provide numbers for twists, diameters, maybe even an image comparison with the downloaded EMDB structure?

We now mention in the text that the diameter of our fibrils was ~ 10 nm, similar to all recombinant and patient-derived structures published to date (lines 109-111): “The fibrils were ~ 10 nm in diameter, similar to recombinant and patient-derived α -Syn fibrils^{33,34} and clearly distinct from neurofilaments (Fig. 1g).”

Approximate values for the twists of the published structures are also listed (lines 138-141): “Our measurements are also consistent with single-particle studies reporting a higher twist, indicative of higher flexibility³⁸, for MSA-derived fibrils³³ (~ 60 nm) than for most structures of recombinant fibrils³⁴ (90-120 nm).”

However, despite extensive image processing efforts, we could not determine an average structure of our fibrils with sufficient resolution, or even reliably calculate their twist, possibly due to their structural variability. Therefore, we would find it premature to present an image comparison at this point.

- Perhaps use of red and green in figures could be prevented? I suggest to change to cyan and magenta throughout the manuscript.

Thank you for the suggestion. We have checked all figures with a color-blind simulator:

<https://www.color-blindness.com/coblis-color-blindness-simulator/>

Furthermore, for all light microscopy images, we show the different channels separately as black and white images.

- L286: The caption reads ‘3D rendering of a)’, but should be ‘3D rendering of tomogram depicted in a)’, as the description for a) reads ‘tomographic slice’, which cannot be 3D-rendered.

Thank you for pointing this out. We have corrected it in all relevant legends (Fig. 1d, Fig. 1h, Fig. 2b, Fig. 3d, Fig. 3e, Supplementary Fig. 6c).

- L304: ‘Cytosolic fibril density’: It is not directly obvious what is meant by this term as it is only explained in the methods section. Perhaps this could be explained in the main text?

We have now added this explanation in the main text (lines 129-131: “However, fibril density within inclusions, defined as the fraction of cytosolic volume occupied by fibrils, was significantly higher in cells expressing GFP- α -Syn (Fig. 2e, Supplementary Table 1).”) and the legend of Fig. 2 (lines 236-237).

- L556: ‘cameras’ -> ‘camera’

“Camera” is used in plural as it refers to both K2 and K3 cameras.

Reviewer #1 (Remarks to the Author):

The authors have answered my previous questions. I have no additional comments.

Reviewer #2 (Remarks to the Author):

The authors have answered all my questions, happy to recommend publication.

Tanmay Bharat